# Viral Infection Leads to a Unique Suite of Allelopathic Chemical Signals in Three Diatom Host–Virus Pairs

**DOI:** 10.3390/md22050228

**Published:** 2024-05-17

**Authors:** Bethanie R. Edwards, Kimberlee Thamatrakoln, Helen F. Fredricks, Kay D. Bidle, Benjamin A. S. Van Mooy

**Affiliations:** 1Department of Marine Chemistry and Geochemistry, Woods Hole Oceanographic Institution, Falmouth, MA 02543, USA; hfredricks@whoi.edu (H.F.F.); bvanmooy@whoi.edu (B.A.S.V.M.); 2Department of Earth and Planetary Science, University of California-Berkeley, Berkeley, CA 94720, USA; 3Department of Marine and Coastal Sciences, Rutgers University, New Brunswick, NJ 08901, USA; thamat@marine.rutgers.edu (K.T.); bidle@marine.rutgers.edu (K.D.B.)

**Keywords:** oxylipins, chemical signaling, lipidomics, diatom viruses, marine virology, allelopathy, viral ecology, lipid, dissolved organic matter, orbitrap mass spectrometry

## Abstract

Ecophysiological stress and the grazing of diatoms are known to elicit the production of chemical defense compounds called oxylipins, which are toxic to a wide range of marine organisms. Here we show that (1) the viral infection and lysis of diatoms resulted in oxylipin production; (2) the suite of compounds produced depended on the diatom host and the infecting virus; and (3) the virus-mediated oxylipidome was distinct, in both magnitude and diversity, from oxylipins produced due to stress associated with the growth phase. We used high-resolution accurate-mass mass spectrometry to observe changes in the dissolved lipidome of diatom cells infected with viruses over 3 to 4 days, compared to diatom cells in exponential, stationary, and decline phases of growth. Three host virus pairs were used as model systems: *Chaetoceros tenuissimus* infected with CtenDNAV; *C. tenuissimus* infected with CtenRNAV; and *Chaetoceros socialis* infected with CsfrRNAV. Several of the compounds that were significantly overproduced during viral infection are known to decrease the reproductive success of copepods and interfere with microzooplankton grazing. Specifically, oxylipins associated with allelopathy towards zooplankton from the 6-, 9-, 11-, and 15-lipogenase (LOX) pathways were significantly more abundant during viral lysis. 9-hydroperoxy hexadecatetraenoic acid was identified as the strongest biomarker for the infection of *Chaetoceros* diatoms. *C. tenuissimus* produced longer, more oxidized oxylipins when lysed by CtenRNAV compared to CtenDNAV. However, CtenDNAV caused a more statistically significant response in the lipidome, producing more oxylipins from known diatom LOX pathways than CtenRNAV. A smaller set of compounds was significantly more abundant in stationary and declining *C. tenuissimus* and *C. socialis* controls. Two allelopathic oxylipins in the 15-LOX pathway and essential fatty acids, arachidonic acid (ARA), eicosapentaenoic acid (EPA), and docosahexaenoic acid (DHA) were more abundant in the stationary phase than during the lysis of *C. socialis*. The host–virus pair comparisons underscore the species-level differences in oxylipin production and the value of screening more host–virus systems. We propose that the viral infection of diatoms elicits chemical defense via oxylipins which deters grazing with downstream trophic and biogeochemical effects.

## 1. Introduction

In marine microbial ecosystems, specific strains of phytoplankton produce oxylipins when stressed, mainly diatom and eukaryotic haptophyte lineages like *Phaeocystis* [1]. Oxylipins are chemical signals that are made by oxidizing fatty acids from membrane lipids. The compounds are used as chemical signals across the tree of life, serving as lipid mediators in the human immune system [2] and natural grazing deterrents for land plants [3]. Oxylipin production by diatoms has been specifically tied to copepod [4] and microzooplankton grazing [5], temperature stress [6], UV stress [7], and nutrient stress [8]. Field studies in the Mediterranean Sea have observed increased oxylipin concentrations in association with cell concentration [9], diatom cell lysis measured by staining for membrane integrity [10], and bloom decline [11]. Over the past two decades, the bioactivity of diatom oxylipins has been extensively investigated by conducting amendment experiments in culture and observing how natural populations respond to high oxylipin concentrations in the field [5,12,13,14,15,16,17,18,19,20]. Taken together this body of work suggests that, as diatoms become stressed and blooms crash, oxylipins have potential to impact microbial community dynamics by deterring microzooplankton grazing [5], causing trophic cascades [20], reducing copepod reproductive success [12], stunting the growth of competitors [13], and stimulating nutrient recycling by particle-associated bacteria [15]. Thus, we sought to understand how another source of stress, viral infection, would impact oxylipin production.

Viruses represent an important mortality term for phytoplankton in the ocean [21,22,23,24,25], and play a role in ocean biogeochemistry shunting carbon from living biomass to the dissolved organic carbon pool [26] and shuttling carbon to depth via sinking particles [21,24,27,28,29,30,31]. Although the effect of viruses on oxylipin production has not yet been characterized for diatoms, studies from higher plants and humans show oxylipin signaling during viral infection [32,33,34,35]. We hypothesized that the viral infection of diatoms would induce oxylipin production, as oxylipin production appears to be a conserved cellular response to a variety of stressors.

Given that the suite of oxylipins produced by diatoms in response to cell wounding is strain specific [1], we sought to explore how different diatom hosts would respond to viral infection and how different types of viruses infecting similar hosts might impact oxylipin production. We conducted lipidomic experiments with three host–virus pairs: *Chaetoceros tenuissimus* infected with a single-stranded DNA (ssDNA) virus, CtenDNAV, *C. tenuissimus* infected with a single-stranded RNA (ssRNA) virus, CtenRNAV, and *Chaetoceros socialis* infected with an ssRNA virus, CsfrRNAV. We found that viral infection led to the enhanced production of several oxylipin species relative to exponential or stationary phase growth and that there was a unique lipidomic signature produced by each diatom host–virus pair.

## 2. Results

### 2.1. Time Course of Infection

Diatom cultures were infected during the exponential phase and incubated alongside uninfected control cultures (Appendix A). Samples for lipidomic analyses were taken at discrete timepoints to capture different phases of growth in the control and different stages of viral infection (Table 1). Uninfected control cultures reached stationary phase on day 4 for *C. tenuissimus* and day 3 for *C. socialis*. A set of follow-up experiments were sampled for lipidomics on day 26 for *C. tenuissimus* and day 21 for *C. socialis*, representing cultures that were in the decline phase. Viral infection by CtenRNAV and CsfrRNAV led to a significant decrease in host abundance on day 3 post-infection; host abundance decreased on day 4 in cultures infected with CtenDNAV (Appendix A). Timepoints from all three of the host–virus experiments were categorized during subsequent supervised statistical analyses based on the stage of infection (early, mid, lysis) or growth phase (T = 0, exponential, stationary, decline) as presented in Table 1.

### 2.2. Defining the Oxylipidome

The LOBSTAHS (Lipid and Oxylipin Biomarker Screening Through Adduct Hierarchy Sequences) lipidomic pipeline was paired with manual peak verification to annotate 156 compounds in the dissolved lipidome as free fatty acids and oxidized free fatty acids (i.e., oxylipins) (Appendix A). An authentic standard retention time model (Appendix A) and an in silico ms^2^ library (Appendix A) were used to obtain more precise annotations of the putative diatom oxylipins (Appendix A). For example, LOBSTAHs annotated a compound as a C16:4 fatty acid with two extra oxygen molecules eluting at a retention time of 1.85 min (FFA 16:4 +2O RT-1.85), but we were able to further annotate the compound as a 9-hydroperoxy hexadecatetraenoic acid (9-HpHTE) based on fragmentation and retention time (Appendix A). To gain a more comprehensive view of the molecular ecology at play, we chose to include all 156 compounds putatively annotated by LOBSTAHs in our statistical analysis, while also highlighting the oxylipins annotated as known allelochemicals produced by diatoms.

### 2.3. Growth Phase and Viral Infection Changes the Dissolved Lipidome

In all three diatom host–virus systems, viral lysis and aging induced the release of free fatty acids and oxylipins in the dissolved organic matter pool (Figure 1). This is true when considering the total peak area of free fatty acids and oxylipins annotated in the lipidome (Figure 1a) and the cell normalized peak area (Figure 1b). The highest overall amount of oxylipins and free fatty acids were produced during lysis in the infected treatments and during early stationary/stationary phase in the control treatments. The amount of oxylipins produced when *C. tenuissimus* was infected with CtenDNAV was significantly higher than in stationary or decline phase controls, but the bulk amount was not distinguishable from early stationary phase. In contrast, *C. socialis* produced a similar amount of free fatty acids and oxylipins during the infection and stationary phases.

Downstream statistical analyses within each diatom host experiment used the peak area data, which are only normalized to the volume filtered and the recovery of the internal standard before mean centering. Each treatment within an experiment started with the same T = 0 and we were ultimately interested in the bulk change in the media. So, we did not use the cell count normalized data, although taking the alternate approach may have yielded even more statistically significant findings, given oxylipin concentration tended to increase as cell counts decreased over time (Figure 1 and Appendix A).

The multivariate techniques, partial least squared differential analysis (PLS-DA) and sparse PLS-DA (sPLD-DA), were used to explore the relative abundances of compounds produced in each treatment with time, as well as pinpoint significant changes in individual compounds. The oxylipidomes of each diatom host–virus pair were analyzed separately. Both lysis and growth phases had a distinct effect on the suite and relative concentration of oxylipins produced (Figure 2a–c). The infection of *C. tenuissimus* with CtenDNAV had the most pronounced impact on the dissolved lipidome with samples clustering further apart in the PLS-DA (Figure 2a) than in other experiments (Figure 2b,c). In the CtenRNAV experiment sPLS-DA, stationary control lipidomes overlapped most with early infection indicating a portion of the stress response to CtenRNAV infection is similar to that of senescence (Figure 2b).

Similarly, the structure of the lipidome during normal growth and CsfrRNAV infection in *C. socialis* was more similar to one another, with the 95% confidence interval of early infection overlapping with stationary and T = 0 (Figure 2c). In each diatom host–virus pair, the dissolved lipidome diverged from the controls as the infections progressed, and the lysis lipidomes were distinct from stationary lipidomes (Figure 2a–c). The lipidomic signature of the decline phase was also easily distinguishable from other control or viral infection timepoints in each experiment (Figure 2a–c; grey ellipses).

### 2.4. Oxylipins and Fatty Acids Associated with Each Diatom Host–Virus Pair

We looked at the significant compounds associated with each multivariate analysis to determine which dissolved compounds were most associated with the lipidomic response to growth phase and viral infection (Figure 2). PLS-DA yielded the best separation for the CtenDNAV infection. Features with variable importance in projection (VIP) scores higher than one for Component 1 (*N* = 54) and Component 2 (*N* = 55) were the most influential in structuring the data (Appendix A). All 156 compounds in the lipidome were considered in the analysis. Component 1 represented 51.8% of the variability in the dataset and Component 2 represented 13% (Figure 2a).

PLS-DA did not adequately separate the treatments and timepoints for CtenRNAV and CsfrRNAV so sPLS-DA was employed to find the most significantly different compounds, with ten compounds assigned to Component 1 and ten compounds to Component 2 (Appendix A). For both CtenRNAV and CsfrRNAV experiments, Component 1 (representing 41% and 45.3% of the variability in the data, respectively) reflected compounds that were associated with viral infection and lysis but were also produced at relatively smaller amounts during the stationary phase (Appendix A). For CtenRNAV experiments, compounds assigned to Component 2 (24.7% of the variability) were upregulated during infection but also significantly associated with the decline phase (Appendix A). For CsfrRNAV experiments, Component 2 (23.2% of the variability) reflected a number of free fatty acids that were released during stationary phase in the controls, as well as oxylipins that were produced during stationary phase but also during decline phase (Appendix A). Notably, compounds assigned to Component 2 were differentially produced during viral infection (Appendix A). Many of the significant oxylipins produced by both *Chaetoceros* species were annotated to the 9-lipoxygenase (9-LOX) C16:n biosynthetic pathways and have known allelopathy which will be discussed in Section 2.5 (Appendix A).

### 2.5. C. tenuissimus Produces a Distinct Suite of Oxylipins Depending on the Infecting Virus

To determine how oxylipin signaling changes when the same host is infected with different viruses, we compared CtenDNAV lysis samples to the CtenRNAV lysis samples (Figure 3). Of the 156 compounds in the dissolved oxylipidome, 20 compounds were significantly more abundant during lysis by CtenDNAV and 26 were significantly more abundant during CtenRNAV lysis (Figure 3a; *t*-tests FDR adjusted *p* < 0.05). A heatmap with the relative abundances of the 46 compounds is used to visualize and quantify these distinctions more clearly (Figure 3b). Ward clustering of the compounds split the lipidomes into two clusters: Cluster 1 are compounds that were more abundant during CtenDNAV Lysis and Cluster 2 are compounds that were more abundant in CtenRNAV Lysis. Parsing the data by cluster and organizing the compounds by the number of carbons in the fatty acid and the number of double bond equivalents revealed a bifurcation in the types of oxylipins elicited by the two viruses (Appendix A). CtenRNAV infection promoted 20:n, 22:6, and 18:3 oxylipin production (Appendix A) whereas CtenDNAV infection resulted in 16:1, 16:2, 16:3, and 18:3 oxylipins being produced (Appendix A). We compared the average chain length, number of double bond equivalents, and number of oxygen moieties for each cluster, and found that Cluster 2—CtenRNAV is comprised of longer chain and more oxidized fatty acids with a similar level of unsaturation compared to the Cluster 1—CtenDNAV (Figure 3c–e).

### 2.6. Known Allelopathic Compounds Produced during Lysis

The above analysis focused on the overall changes in the lipidome, with many compounds not having known bioactivities (allelopathy, stimulation, stress surveillance, etc.) in marine diatom ecosystems. We further parsed the data, looking only at oxylipins in the 20:5, 20:4, and 16:n oxylipin pathways that have been well described in diatoms. We iterated upon the 5-, 6-, 8-, 9-, 11-, 14-, and 15-LOX pathways documented in the review by Andreou et al. 2009 [36], building an in silico database to identify potentially allelopathic oxylipins (Appendix A; Appendix A).

As previously mentioned in the overall analysis of the dissolved lipidome (Section 2.2), viral infection induced the production of several compounds from the 9-LOX pathway with evidence of 6-LOX, 11-LOX, and 15-LOX pathways being triggered as well (Appendix A). Many of the 9- and 15-LOX compounds produced over the course of infection are known to decrease copepod fecundity in the Mediterranean Sea, and are allelopathic towards dinoflagellates, copepods, and sea urchins in laboratory studies [5,11,18,37,38] (Table 2). The other compounds from these LOX pathways are hypothesized to be allelopathic based on homology (Figure 4 and Figure 5).

Thirteen features positively annotated as homologs to allelopathic compounds were significantly more abundant during viral lysis by CtenDNAV relative to control treatments (Figure 4a–n; ANOVA, Fisher’s LSD (least significant difference) post-hoc *p*-value << 0.05 with FDR correction). Likewise, twelve allelopathic homologs were significantly ‘upregulated’ in the CtenRNAV experiment and seven in the CsfrRNAV experiment (Figure 4a,c–n and Figure 5a–i). While the total peak area of the oxylipidome was not significantly larger in CtenDNAV- vs. CtenRNAV-infected treatments (Figure 1b), most of the allelopathic compounds were more abundant during lysis by CtenDNAV (Figure 4a–n). Every compound presented was significantly more abundant during CtenDNAV and CtenRNAV lysis compared to control treatments, except for 6-oxo hexadecaenoic acid (6-oxoHME; Figure 4b) and 8-hydroxy eicosatetraenoic acid (8-HETE; Figure 4j). 6-oxoHME was not significantly more abundant during CtenDNAV or CtenRNAV lysis than in the decline control treatments but was significantly upregulated during lysis by CsfrRNAV (Figure 5a), whereas 8-HETE was significantly upregulated during viral lysis with CtenDNAV but not CtenRNAV or CsfrRNAV (Figure 4j).

Two compounds belonging to the 15-LOX biosynthesis pathway, 15-hydroperoxy eicosapentaenoic acid (15-HpEPE) and 15-oxo eicosadienoic acid (15-oxoEDE), were more abundant during the stationary phase in *C. socialis* compared to during lysis via CsfrRNAV (Figure 5h,i). However, other 15-LOX compounds, 15-hydroxy eicosatrienoic acid (15-HETrE) and 11,14-dihydroxy eicosapentaenoic acid (11,14-diHEPE; Figure 5e,g), show the typical pattern with a higher abundance in CsfrRNAV lysis samples compared to *C. socialis* stationary or decline samples, suggesting cellular conditions target the downstream regulation of the specific oxylipin species produced by LOX pathways. A diagram outlining these diverging biosynthesis pathways can be found in Appendix A. In our experiments, stationary phase *C. socialis* favored 15-LOX acting on the fatty acid EPA, resulting in 15-HpEPE production, whereas, during viral lysis the 15-LOX EPA pathway elicited an unknown enzyme that produced 11,14-diHEPE. During stationary phase, 15-LOX also acted on C20:3 fatty acids with subsequent allene oxide synthase activity, producing 15-oxoEDE, whereas CsfrRNAV lysis led to peroxidase activity within the 15-LOX ETrE pathway and the production of 15-HETrE instead.

### 2.7. Comparing the Diatom Host–Virus Pairs

Of the 156 oxylipins and fatty acids in the dissolved lipidome, twenty-four were more abundant during viral lysis in all three diatom host–virus experiments compared to stationary growth or decline (Figure 6). Six known allelopathic compounds, 6-oxoHME, 9-hydroperoxy hexadecatrienoic acid (9-HpHTrE), 9-hydroxy hexadecatetraenoic acid (9-HHTE), 9-HpHTE, 9-HETE, and 11,14-diHETE, are amongst the 24 oxylipins making up the core oxylipidomic response to viral infection in *Chaetoceros* diatoms (Table 3). The lipidomes from the experiments infecting *C. tenuissimus* with either CtenDNAV or CtenRNAV shared seventy-eight oxylipins that were ‘upregulated’ during viral lysis relative to stationary or decline phases. Eleven oxylipins were uniquely associated with CtenDNAV lysis, fourteen oxylipins were uniquely associated with CtenRNAV lysis, and one single compound was unique to CsfrRNAV lysis (Appendix A). This intercomparison of oxylipins associated with lysis across all three experiments further suggests that the diversity of oxylipins produced depends on the type of virus infecting and the specific diatom host being infected.

To integrate across the experiments in a more quantitative manner, the program MINT (Multivariate INTegrative method) from the mixOmics package [43,44] was used to collectively analyze all of the data with sPLS-DA, which removes compounds that did not contribute significantly to the structuring of the data and assigns as many compounds to each component as necessary to reduce classification error (Figure 7 and Figure 8). MINT was developed to integrate across independent meta-omic experiments and identify reproducible molecular signatures. While we noted differences in the magnitude and timing of allelopathic compound production in host–virus pairs (Figure 1, Figure 2, Figure 3, Figure 4 and Figure 5) and the shared and unique compounds associate with lysis (Figure 6), we used MINT to identify the most conserved lipidomic response to viral infection and growth phase across the three experiments (Figure 7 and Figure 8). We ran the analysis twice. First, we used a proxy for the absolute abundance of the compounds, peak area, in order to assess the most abundant and distinct compounds associated with each treatment and timepoint (Figure 7). The second analysis used the relative abundance of the compounds, giving equal weight to high abundance and low abundance compounds within the dataset (Figure 8).

When considering the absolute abundance of all of the compounds in the dissolved oxylipidome, X-variate 1, representing 43% of the explained variability, separated the samples by the progression of viral infection and X-variate 2 (13% of the variability) picked up on similarities between the stationary phase and viral infection (Figure 7a). The one compound that was most significantly correlated with X-variate 1 was 9-HpHTE (FFA 16:4 +2O RT-1.85), which had an increasingly positive correlation with early infection, mid-infection, and lysis (Appendix A) and exhibited higher concentrations as infection progressed (Figure 7b,c). X-variate 2 had more significant compounds, but they were mostly free fatty acids and oxylipins not known to be allelopathic in diatoms (Appendix A). Specifically, the essential free fatty acids, arachidonic acid (ARA; FFA 20:4 RT—8.94), eicosapentaenoic acid (EPA; FFA 20:5 RT—8.43), and docosahexaenoic acid (DHA; FFA 22:6 RT—8.89) were most significantly correlated with the stationary phase (Appendix A), a trend driven by the *C. socialis* experiments (Figure 7e,g,i) but also seen in *C. tenuissimus* controls (Figure 7d,f,h). Interestingly, these compounds were produced at a higher concentration during the lysis of *C. tenuissimus* with CtenRNAV and early on during *C. socialis* infection with CsfrRNAV but not during the lysis of *C. tenuissimus* by CtenDNAV (Figure 7d–i).

A different suite of conserved oxylipins associated with the growth phase and viral infection were highlighted when considering the MINT sPLS-DA of relative abundance data (Figure 8). X-variate 1 (36% explained variability) still separated the lipidomes based on the progression of viral infection but now X-variate 2 (17%) reflected the lipidomic signature of the decline phase (Figure 8a). The allelopathic oxylipin 11,14-diHEPE along with two novel C14:2 +1O oxylipins were significantly associated with X-variate 1 and were positively correlated with infection and lysis (Appendix A). Relative to the mean, these compounds were not abundant in the control treatments but increased in abundance as viral infection progressed in all three diatom host–virus pairs (Figure 8b–g). Four other allelopathic compounds, 5-epoxy hexadecaenoic acid (5-epHME), 9-HHME, 9-oxo hexadecadienoic acid (9-oxoHDE), and 15-hydroxy eicosapentaenoic acid (15-HEPE), were significant for X-variate 2 and were most positively associated with the decline phase (Appendix A). These four allelopathic compounds were also produced during viral infection but the sPLS-DA picked up on the concomitant increase in the decline phase as being diagnostic (Figure 8h–o). This analysis indicates that while oxylipin production during lysis is distinct from decline, these specific compounds are released as chemical signals regardless of the two modes of diatom mortality tested here (i.e., senescence vs. viral lysis).

## 3. Discussion

### 3.1. Oxylipin Production Was Stimulated by Viral Infection and Distinct from the Lipidomic Signature of the Growth Phase

Diatoms are important primary producers in marine settings that produce allelopathic oxylipins when physiologically stressed or wounded by grazing [4,5,6,7,8,45]. Here, we present the first evidence that diatoms also produce these compounds in response to viral infection and the suite of compounds produced likely influence trophic transfer within the microbial loop. The viral infection of the cosmopolitan diatoms *C. socialis* and *C. tenuissimus* resulted in an enhanced production of oxylipins relative to control treatments (Figure 1, Figure 3, Figure 4 and Appendix A). *C. tenuissimus* infected with CtenDNAV resulted in the most distinct lipidomic response, as evidenced by the ease with which PLS-DA distinguished the various lipidomes from different timepoints and treatments (Figure 2a). While the overall oxylipin profile during viral lysis was unique from the oxylipin profile of the controls in all three diatom host–virus pairs tested, the control lipidomes overlapped with early infection in both ssRNA virus treatments (Figure 2b,c). For the majority of compounds identified via multivariate analysis as being significant, virus-infected treatments exhibited the highest relative concentrations (Figure 2, Appendix A).

Many of the identified oxylipins are known to be bioactive in marine ecosystems, functioning as allelopathic compounds towards grazers (Figure 3, Figure 4 and Appendix A; Table 2). Specifically, these same compounds were observed in *Skeletonema marinoi* [39], *Thalassiosira rotula* [40], *Pseudo-nitzschia delicatissima* [41], *S. pseudocostatum* [37], and *S. costatum* [39] cultures and were found to decrease copepod reproductive success in collapsing phytoplankton blooms in the Mediterranean Sea [18]. We observed nine different 9-LOX oxylipins that were upregulated across the various viral infection experiments (Figure 4 and Figure 5; Appendix A), with 9-HpHTE being identified as the best biomarker for viral infection when considering the absolute abundance of oxylipins across the lipidome (Figure 7b,c). We also observed evidence of 6-LOX, 11-LOX, and 15-LOX oxylipin biosynthesis pathways being associated with viral infection (Figure 4 and Figure 5). Previous oxylipin screening in *C. socialis* and *C. affinis* revealed the production of 14-LOX and 8-LOX oxylipins from the C20:5 pathways [39]. Differences in oxylipin production between species of *Chaetoceros* are consistent with Wichard et al. (2005) [1], where fifty-one species of phytoplankton were screened and the production of PUA oxylipins was found to be species specific.

#### 3.1.1. Lysis by ssRNA Viruses Results in Different Oxylipins Than ssDNA Viruses

The type of virus infecting the diatom host also appears to be important for oxylipin production; forty-six differentially “expressed” oxylipins were detected when *C. tenuissimus* was lysed by CtenDNAV vs. CtenRNAV (Figure 3a,b). The lysis of *C. tenuissimus* by CtenRNAV was associated with oxylipins made of longer chain fatty acids and more oxidations (Figure 3c–e and Appendix A). In experiments with the different chain lengths of oxylipins called polyunsaturated aldehydes (PUAs), longer chain PUAs (e.g., C10:2 and C10:3) were toxic to copepods, bacteria, phytoplankton, and sea urchin larvae at lower concentrations than shorter chain PUAs (e.g., C7:2 and C8:2) [12,13,14,38]. Therefore, it follows that the chemical signaling potential of dissolved organic matter produced in response to CtenRNAV may be greater than that produced in response to CtenDNAV. Conversely, among the known allelopathic compounds in this dataset, concentrations were higher in the CtenDNAV lysis timepoint (Figure 3 and Figure 4). The oxylipin inhibition of microzooplankton, microalgae, bacterioplankton, and particle-associated communities is dose-dependent so it is difficult to postulate which virus would contribute the most to chemical signaling within the surface ocean and whether the response of the community would be inhibition or stimulation. We explore the bioactivity and biogeochemistry of virus-induced oxylipin production further in Section 3.4.

The differences in lipidomic response to ssDNA vs. ssRNA viruses raise intriguing questions about molecular level differences in the progression of infection that could alter the quality and bioactivity of organic matter released into the dissolved pool. For example, AA, EPA, and DHA synthesis are compartmentalized to the endoplasmic reticulum (ER), suggesting that ssRNA viruses disrupt and oxidize the ER [46]. Relatively shorter chain PUFAs (C16:n and C18:n) are synthesized in the chloroplast, suggesting that ssDNA viruses may cause more oxidative stress in the chloroplast [46]. The ER is more protein rich than the chloroplast. Experiments growing bacteria on the intra and extracellular organic material from the *C. tenuissimus* ssDNA/ssRNA system show that ssRNA virus stimulates exoproteolytic activity of bacteria, whereas ssDNA virus does not [47], which is consistent with ssRNA disrupting the ER and releasing protein-rich DOM. Subcellular compartmentalization experiments would need to be done to test these hypotheses (Figure 9).

Oxylipin production was also associated with the growth phase. While viral infection and lysis often led to higher concentrations of many oxylipins, there was a subset of compounds that were significantly more abundant in stationary and decline phases relative to lysis, specifically during the infection of *C. socialis* with CsfrRNAV (Figure 5h,i, Appendix A). This is consistent with previous studies suggesting that oxylipin production is a normal part of the diatom secondary metabolome as cells near senescence [45] and field observations of oxylipins in the surface waters where blooms are collapsing [9,10,11,48]. However, our data suggest that bloom decline due to viral infection would produce higher concentrations of oxylipins in the surrounding waters than blooms terminated due to nutrient limitation alone, and certain species of diatoms and virocells may be more prone to oxylipin signaling than others.

#### 3.1.2. Relative Quantification and the Dose-Dependent Bioactivity of Oxylipins

Again, oxylipin bioactivity is dose dependent. For example, PUAs stimulated the respiration and enzymatic activity of particle-associated communities at 1–10 µM in amendment experiments but inhibited these communities at 100 µM in the North Atlantic [15]. Mesocosms adding 1.25 µM PUAs every day to natural plankton assemblages from the Bothian Sea (Sweden) showed no impact on community composition or bacterial cell abundance, whereas mesocosms seeded with oxylipin producing *Skeletonema* had maximum PUA concentrations of 1–6 nM but showed a shift towards Chrysophytes and Euglenophytes and a decrease in bacterial cell abundance [49]. *Skeletonema* is known to make oxylipins upstream of PUAs (Table 2), and such compounds may have been playing a role in the overall response of the community. It should be noted that we did not observe any PUAs during our laboratory experiments with *Chaetoceros* spp. Rather, we observed the longer chain oxylipins, which are upstream of PUAs and are known to be more detrimental to zooplankton than PUAs, causing copepod reproductive success failures and altering *Oxyrrhis marina* grazing at lower concentrations than PUAs [5,18].

The differential response of similar taxa to oxylipins, the differential response of organisms to various oxylipin chain lengths, and the dose dependence of the magnitude and sign of the response are all prime reasons the absolute quantification of oxylipins in the future is necessary. We presented an assessment of the relative abundance and peak area of a wide variety of oxylipins from laboratory culture experiments; we did not attempt to do absolute quantification when we analyzed the fresh-underivatized samples in 2016. Using the peak area of the 10 µM internal standard (~10^6^), we estimate that the concentration range in our experiments is orders of magnitude higher (total oxylipidome peak area in CtenDNAV ~5 × 10^8^; Figure 1a). Keeping in mind that in situ *Chaetoceros* blooms are typically 3 × 10^4^ cells per mL and our experiments were 5 × 10^5^–4.5 × 10^6^ cells per mL (Appendix A), we can extrapolate to a natural viral infection scenario, estimating concentrations on the order of 1–10 µM. However, quantitative studies conducted in the Mediterranean show maximum cell-associated oxylipin concentrations in the nM range [9], orders of magnitude lower than our rough estimate, but high enough to influence grazing [5], again underscoring the importance of making more measurements in the environment.

### 3.2. Novel Oxylipins

We annotated one C20:5 +2O oxylipin as an 11,14-dihydroxy acid, a subclass of oxylipins that have not been previously described in radiotracer experiments with diatom cultures but have been observed in the marine environment and in lipidomic studies with *Phaeodactylum tricornutum* [5,11]. Higher plants are also known to produce dihydroxy acids via alpha-dioxygenase [50,51]. Hydroxyepoxy acids (Hep) and hydroperoxy acids (Hp) are the typical oxylipins with two oxygens (+2O) produced by diatoms [36]. The observed retention times were too early to be either of those types of oxylipins and matched RT predicted by authentic dihydroxy acid standards for C20:4 analogs as well as ms^2^ fragmentation predicted by an in silico model [52] (Appendix A; Appendix A).

We previously evaluated the allelopathic potential of commercially available dihydroxy acids that function as chemical signals in mammalian immune systems and found that they decreased the microzooplankton grazing of the dinoflagellate *Oxyrrhis marina* on the diatom *P. tricornutum* [5]. The chemical signaling potential of dihydroxy acids towards other surface ocean microbes inhabiting the phycosphere must be investigated to fully understand the biogeochemical implication of virally induced oxylipin signaling. Furthermore, these oxylipins may be commercially valuable as reviews by Yi et al. and Rucco et al. highlighted studies showing the anti-inflammatory, anti-cancer, and antimicrobial potential of diatom-derived oxylipins [53,54].

The combined multivariate analysis also highlighted novel 14:2 +1O oxylipins, and a whole range of C18 oxylipins that are typically associated with cyanobacteria and fungi [36] and C22 oxylipins associated with *Lepidocylindrus* diatoms [55], as important oxylipins (Appendix A). These are good candidates for future amendment experiments as we know little about their function in marine diatom ecosystems. As mentioned above, amendment experiments showed that some organisms are stimulated by oxylipins, like the bacterial epibionts of diatoms [14] and particle-associated communities [15,56], but most organisms are inhibited in some way by PUAs, including free-living bacteria and particle-associated communities at higher concentration exposures [5,13,14,19,57].

### 3.3. Potential Biomarkers for Viral Infection

Lipids have been used as biomarkers to survey the geological past and modern biogeochemical processes [58]. For example, glycosphingolipid biomarkers are used to detect and diagnose viral infection of the cosmopolitan coccolithophore, *Emiliania huxleyi*, and an increase in fatty acid unsaturation has been proposed as a biomarker for cyanophage infection [59,60]. Here the lipidomes of all diatom host–virus pairs were analyzed together using discriminate analysis techniques to identify biomarkers for each treatment (Figure 7 and Figure 8). Allelopathic 11,14-diHEPE and 9-HpHTE, along with novel 14:2 +1O oxylipins were most correlated with viral infection (Figure 7b, Figure 8b–d, Appendix A) and may prove to be useful biomarkers of viral infection in situ. Given the species level differences in oxylipin response, other diatom host–virus pairs should be screened to see how widespread this lipidomic response is across diatom lineages or whether it is specific to *Chaetoceros*.

Other allelopathic compounds, 6-oxoHME, 9-oxoHDE, and 9-HHTrE, and 15-HEPE, were upregulated in some viral treatments but were also significantly associated with the decline phase relative to other timepoints, indicating an overlap in chemical signaling as cells become physiologically stressed during aging (Figure 4, Figure 8h–o and Appendix A). This complicates the search for oxylipin biomarkers of viral infection as many compounds are associated with compromised growth phases (Figure 5h,i, Figure 7, Figure 8, Appendix A). In this regard, oxylipins may best serve as general biomarker for phytoplankton stress, as they are in the observations from the Mediterranean Sea [10]. Lastly, saturated long chain (C18–C20) and very long chain (C22–C26) free fatty acids are relatively rare fatty acids that are chemotaxonomic for cyanobacteria and eustigmatophytes [61] but were associated with CtenRNAV infection here (Figure 3b,c) and should be explored as a diagnostic tool in the future [57].

### 3.4. Ecological Implications

#### 3.4.1. Allelopathy

The laboratory-based findings presented here suggest that the viral infection of *Chaetoceros* populations, a globally dominant genus [62], would induce specific chemical cues in natural environments with microbial ecosystem consequences. A unique feedback loop may exist between two modes of phytoplankton mortality, whereby viral infection within a community may decrease grazing pressure by microzooplankton immediately and depress copepod population size on longer time scales. This would prolong *Chaetoceros* blooms by decreasing one mode of mortality, while the other modes continue. This may also be advantageous to viruses by preserving the host population while infection has time to proceed and by increasing encounter rates post lysis.

#### 3.4.2. Dissolved Organic Matter Quality

Omega-3 and omega-6 fatty acids are nutritionally important for higher trophic levels and are primarily produced by marine microalgae [63]. We observed that the stationary controls were releasing more unoxidized omega-3 fatty acids into the dissolved phase than in the CtenDNAV-infected treatments (Figure 7f,h). Similarly, the CsfrRNAV infection of *C. socialis* decreased the amount of omega-3 and omega-6 fatty acids compared to the stationary control (Figure 7e,g,i). However, during the viral infection of *C. tenuissimus* with CtenRNAV, EPA, DHA, and AA were more abundant than during the stationary phase in the control (Figure 7d,f,h). It should be restated that *C. socialis* made an order of magnitude more ARA, EPA, and DHA than *C. tenuissimus* experiments (Figure 7d–i).

Therefore, the nutritional value of DOM released with respect to essential fatty acids varies across diatom host–virus pairs and can be expected to have an impact on surface ocean biogeochemistry mediated by osmotrophic organisms like bacterioplankton, marine fungi/saprophytes, and mixotrophs. We discuss findings from members of our team that support this idea in Section 3.1.1 above, showing bacterial exoproteolytic activity was higher when they were exposed to organic matter from CtenRNAV infections compared to CtenDNAV infections. Furthermore, *C. tenuissimus* made more allelopathic oxylipins during infection with an ssDNA virus compared to an ssRNA virus (Figure 3 and Figure 4), which may be having an inhibitory effect on the bacteria tested.

#### 3.4.3. Probability of Oxylipin Signaling in the Environment

While the exact bioactivity of the mix of oxylipins in the lysates remains to be seen, the extent of oxylipin signaling in the natural environment will depend on the viral and phytoplankton community composition. Meta-transcriptomic analyses of the global TARA dataset suggests that RNA viruses mostly infect eukaryotes [64]. However, it is unclear how much of global phytoplankton mortality is attributable to ssDNAV vs. ssRNAV [65]. There is a long history of methodological challenges to enumerating the major types of viruses found in the marine environment (dsDNA, ssRNA, dsRNA, and ssDNA) in a quantitative and intercomparable way [66]. However, several lines of evidence support that oxylipin signaling during viral infection is likely a common occurrence in the global ocean. *Chaetoceros* is the most abundant genus of diatoms in the ocean, representing wide diversity that is well suited to adapt to climate change [67]. CtenDNAV and CtenRNAV sequences were associated with *Chaetoceros* cells found in the sediments of the Yatsushiro Sea, Japan and CtenRNAV and other ssRNA diatom viruses were associated with bloom decline and silicon stress across the Pacific in the California Current Ecosystem [23,68]. Lastly, lipoxygenase (LOX) sequences correlate with the relative abundance of diatoms in the global TARA Oceans dataset [9].

#### 3.4.4. Impacts of Oxylipins on Ocean Biogeochemistry

The impact of viral infection on the oceanic carbon cycle has been conceptualized as a viral shunt whereby viral-mediated lysis moves carbon from the particulate organic pool to the dissolved organic pool where it can be respired by bacteria in the surface ocean [26] or as a viral shuttle, whereby viral infection induces host responses that stimulate aggregation and facilitate sinking [27,28,69]. Here we present an additional mechanism whereby viruses may alter carbon cycling by eliciting the production of chemical signals that decrease trophic transfer and export to depth by transport mechanisms associated with grazers (diel vertical migration, seasonal lipid pump, and fecal pellet production) and likely alters the bacterial remineralization of diatom-derived organic matter as well.

Oxylipins may also be involved in the aggregation response to viral infection, as oxylipins have been shown to increase TEP production in mesocosm experiments [70]. Furthermore, Yamada et al. found that Coomassie-stainable particles (CSP) and large particle concentrations increased in CtenDNAV infections compared to *C. tenuissimus* controls over a 2-week laboratory time-series, but TEP did not increase over time [27]. The impact of oxylipins on diatom aggregation and CSP production has not yet been explored. CSP contains more protein-rich exudates and CSP staining could be used to determine whether CtenRNAV releases more protein-rich organic matter compared to CtenDNAV as suggested by this study and Kranzler et al. (in prep) [47]. More mesocosm experiments, laboratory experiments with other host–virus systems, and in situ process studies are needed to determine the net effect of oxylipin chemical signaling triggered by viral infection on carbon cycling and whether the predicted impacts on trophic dynamics and carbon cycling occur outside of the lab.

## 4. Materials and Methods

### 4.1. Culture Experiments Infecting Diatoms with Viruses

Three viral infection experiments were set up. Triplicate cultures of *Chaetoceros socialis* stain L-4 were infected with an ssRNA virus (referred to as CsfrRNAV) and triplicate cultures of *C. tenuissimus* strain 2–10 were infected with an ssRNA virus and an ssDNA virus separately (referred to as CtenRNAV and CtenDNAV, respectively) [71,72,73]. Triplicate control cultures of uninfected diatoms were also set up in parallel to the infected treatments for each experiment. All cultures were grown in SW-3 media at 15 °C and 150 μE m2 d light intensity. The dissolved lipidome was sampled along the course of infection using solid phase extraction. At each timepoint, 30 mL of each treatment was pre-filtered through a 0.2 µm Durapore filter (EMD Millipore, Burlington, MA, USA) to remove particles. The deuterated internal standard, 15(S)-Hydroxy eicosatetraenoic acid-d8 (Cayman Chemical, Ann Arbor, MI, USA), was added to each filtered sample to a final concentration of 10 µM. The dissolved lipids were then extracted onto an HLB-SPE cartridge (Waters, Milford, MA, USA) and flash frozen in liquid nitrogen. Throughout the experiment, samples were also taken for cell counts to monitor the course of infection using microscopy, and for the control decline experiments, flow cytometry.

### 4.2. Analysis of the Dissolved Lipidome Samples

The dissolved lipids were eluted from the SPE cartridge with 2 mL of 70:30 acetonitrile: isopropanol into pre-combusted collection vials that were treated with the antioxidant BHT to prevent autoxidation of the samples. The samples were immediately transferred from the collection vials into HPLC vials and capped under argon. The samples were stored at −80 °C until mass spectrometric analysis. Within a week of extraction, the dissolved lipidome samples were analyzed using reverse-phase HPLC (Agilent 1200 system; Agilent, Santa Clara, CA, USA) paired with high-resolution, accurate-mass (HRAM) data from a Thermo Exactive Plus Orbitrap mass spectrometer (ThermoFisher Scientific, Waltham, MA, USA). The chromatographic method used a Xbridge C8 column (Waters, Milford, MA, USA) as the stationary phase, 18 MΩ water as eluent A, 70:30 acetonitrile: isopropanol as eluent B, and ammonium acetate and acetic acid as the adduct-forming additives [74]. The gradient began with a 1-min isocratic hold of 45% B and shifted to 99% B over 25 min with a flow rate of 0.4 mL min^−1^. Full scan data (*m*/*z* 100–1500) were collected in negative mode at a mass resolution of 170,000.

### 4.3. Lipidomic Workflow

The raw mass spectrometric data were converted to mzXML format with msconvert and analyzed with the R packages xcms [75,76] and CAMERA [77], which aligned the chromatograms, picked and integrated peaks, and identified and removed secondary isotopic peaks. The lipidomic features were further annotated using the LOBSTAHS package which included *m*/*z* information of the various acyl chain-lengths, degrees of unsaturation, and oxidations of triacylglycerols, free fatty acids (FFA), polyunsaturated aldehydes (PUA), and eight intact polar diacylglycerol lipids and their most common adducts [42]. The features annotated as oxylipins and free fatty acids were retained and manually verified as high-quality peaks in MAVEN [78,79]. Peak areas were normalized to the recovery of the internal standard and volume filtered. Statistical analyses were performed using the R package MetaboAnalyst through its web interface [80] and the mixOmics package in R [43,44]. Please see Appendix A for a breakdown of the lipidomic intercomparisons during statistical analysis.

Annotation to the functional group level was first achieved by comparing the observed retention times (RTobs) to RT predicted from a set of 18 authentic standards (Appendix A) [5]. Four years after the initial analysis, the samples were reanalyzed on a UPLC system paired to a Thermo Orbitrap ID-X instrument which allowed for selective ms^n^ fragmentation with the goal of verifying the functional group annotations and gaining positional information on features putatively annotated as known allelopathic oxylipins. Chromatography was completed using an Accucore C8 column (ThermoFisher, Waltham, MA, USA) as the stationary phase and the same eluents, additives, and gradient as the initial analysis. To get better separation and isolation for ms^2^, CsfrRNAV samples were also analyzed with a modified gradient that started at 20% B and increased to 99% B over 30 min.

An oxylipin fragmentation database was built using Competitive Fragmentation Modeling-ID (CFM-ID) to model the fragmentation of oxylipins produced by diatoms as well as oxylipins predicted to exist based on the known lipoxygenase enzymes [81], (Appendix A). MS-DIAL was used to compare the observed ms^2^ fragments of the proposed allelopathic oxylipins in the dataset to the reference database [82] (Appendix A and Appendix A. A new set of authentic standards was also run with ms^2^ fragmentation and compared to a CFMID model to assess the range of acceptable dot products, reverse dot products, and presence scores computed by MSDIAL (Appendix A; Appendix A; Appendix A). Guidelines set forth by Chemical Analysis Working Group (CAWG) Metabolomics Standards Initiative (MSI) [83] require a retention time match to an authentic standard, exact mass information, and characteristic fragmentation for a feature to be considered a high-confidence annotation. Allelopathic compounds annotated to the positional level met those guidelines (Figure 3 and Figure 4) by matching ms^2^ fragmentation to in silico reference standards and using calculated retention times from analogs of different carbon chain length and double bond equivalents. Annotations in the LOBSTAHS format (carbon number: double bond equivalent + oxygenation + retention time) meet the exact mass criteria. Please see Appendix A for an overview of the lipidome annotation pipeline used. Note there are many methods in the literature for measuring oxylipins that include different types of mass spectrometry and chromatography, various derivatization schemes, and alternate annotation pipelines [84,85,86,87,88].

## 5. Conclusions

Viral infection led to an increased production of various oxylipin in two species of diatoms belonging to the genus *Chaetoceros.* Both ssRNA and ssDNA viruses elicited a distinct change in the dissolved lipidome compared to uninfected host controls over the course of infection, culminating in lysis. The oxylipins released in the dissolved organic matter pool during the infection of *C. tenuissimus* with CtenDNAV was different from those released during lysis with CtenRNAV, underscoring the importance of exploring species-specific oxylipin signaling and considering community composition when studying oxylipins in situ. Allelopathic compounds known to impact microzooplankton grazing and copepod reproduction were significantly upregulated in all virus infection treatments, along with several compounds with unknown bioactivities in the surface ocean ecosystem. We further hypothesize that oxylipin production in response to viral infection may decrease grazing pressure on phytoplankton communities and alter the flow of carbon through the system.

## Figures and Tables

**Figure 1 marinedrugs-22-00228-f001:**
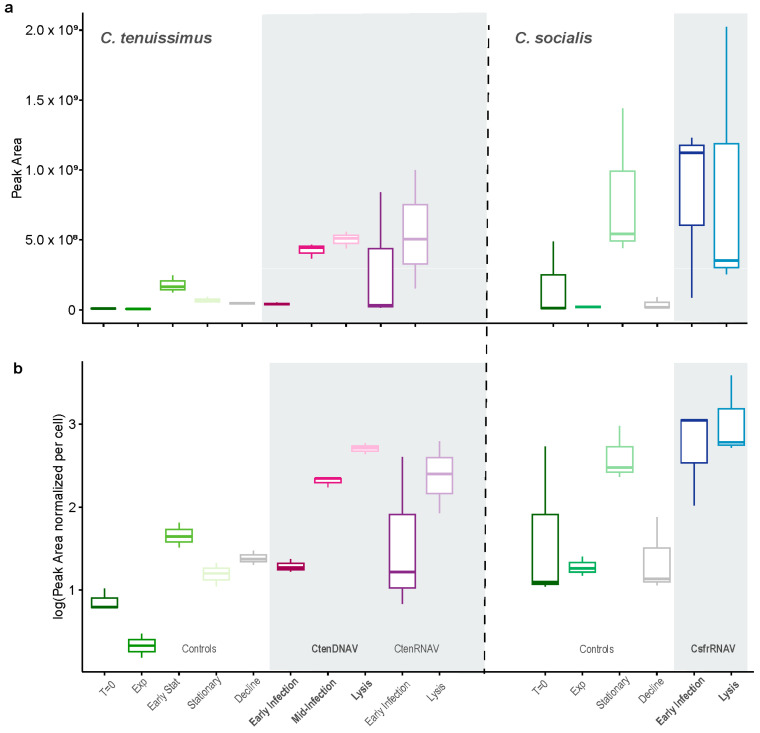
Viral infection and senescence increase the amount of free fatty acids and oxylipins released into the dissolved organic matter pool by *Chaetoceros* spp. (**a**) Average summed peak area of features annotated as free fatty acids or oxylipins within the dissolved lipidome for each experimental timepoint (*N* = 3 for each). (**b**) Summed peak area of free fatty acids and oxylipins within the dissolved lipidome for each experimental timepoint normalized to cell count and log transformed (*N* = 3). Analysis of variance (ANOVA) with a Tukey HSD (honestly significant difference) post-hoc test was run on the cell normalized data (Appendix A). CtenDNAV lysis, CtenRNAV lysis, and CsfrRNAV lysis timepoints were not statistically distinct from one another, nor was *C. socialis* stationary phase statistically distinct from *C. tenuissimus* stationary or early stationary phase. Lysis timepoints were also not statistically distinct from the early stationary phase control in the CtenDNAV and CtenRNAV experiments. However, CtenDNAV lysis and mid-infection timepoints were statistically higher than exponential, stationary, and decline phases as well as early infection. CsfrRNAV Lysis timepoints were statistically distinct from *C. socialis* decline but not stationary timepoints.

**Figure 2 marinedrugs-22-00228-f002:**
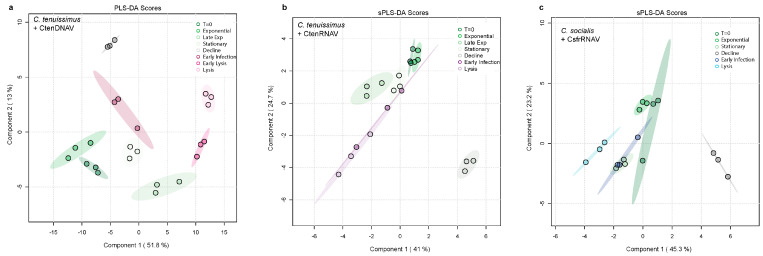
Viral infection induces large scale changes in the dissolved lipidome of *Chaetoceros* spp. and the release of free fatty acids and oxylipins. This response is distinct from what is observed as cultures age. (**a**–**c**) Multivariate analyses of the dissolved lipidomes at T = 0, exponential phase, stationary phase, and decline phase controls (green), and early, mid, and late infection in the virally infected treatments (pink—CtenDNAV, purple—CtenRNAV, blue—CsfrRNAV). Ellipses denote the 95% confidence interval for each timepoint. (**a**) PLS-DA loadings of samples from experiment infecting *Chaetoceros tenuissimus* 2–10 with CtenDNAV showing that each lipidome sampled was statistically distinct, (**b**) sparse PLS-DA loadings of lipidomes from *C. tenuissimus* 2–10 infected with CtenRNAV, and (**c**) *Chaetoceros socialis* infected with CsfrRNAV distinguished between the lipidomes better than PLS-DA with lysis and decline timepoints exhibiting a distinct signal from early infection, T = 0, and exponential phase.

**Figure 3 marinedrugs-22-00228-f003:**
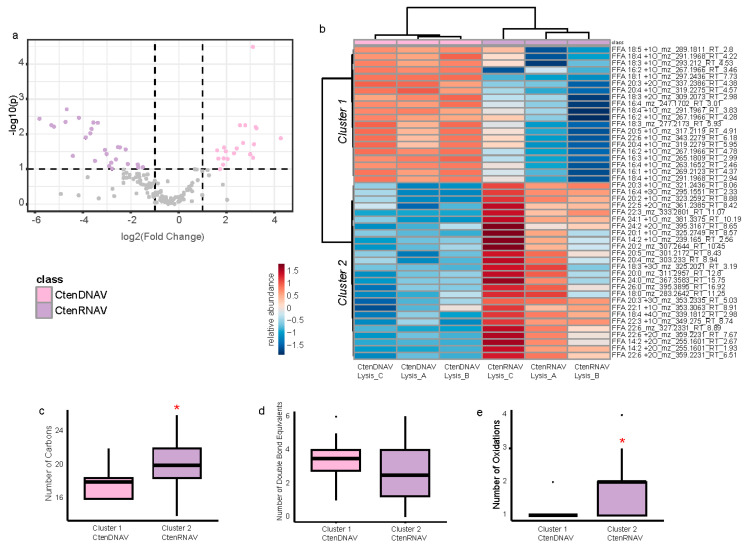
A comparison of experiments infecting *Chaetoceros tenuissimus* with either CtenDNAV or CtenRNAV shows that a different suite of oxylipins is produced depending on the infecting virus. (**a**) Volcano plot showing the log fold change in relative abundance of each compound vs. the log *p*-value from the ANOVA. Compounds that were significantly more abundant during lysis in the CtenDNAV are highlighted in light pink (*N* = 20) and compounds that were significantly more abundant in the CtenRNAV experiments are highlighted in lilac (*N* = 26). Compounds in grey were not significantly different between treatments. (**b**) Heatmap showing the log2 fold change in the abundance of each compound relative to the mean abundance across all samples. Red indicates an increase relative to the mean and blue indicates a decrease. Ward clustering grouped the samples by treatment and the compounds into Cluster 1, compounds that were more abundant in the CtenDNAV experiments, and Cluster 2, compounds that were more abundant in the CtenRNAV experiments. (**c**) Average chain length, (**d**) double bond equivalent, and (**e**) level of oxidation of all compounds in Cluster 1 (associated with CtenDNAV, pink) and Cluster 2 (associated with CtenRNAV, purple). Asterisks denote significant difference between the clusters by ANOVA and Tukey HSD post-hoc test (*p* < 0.05).

**Figure 4 marinedrugs-22-00228-f004:**
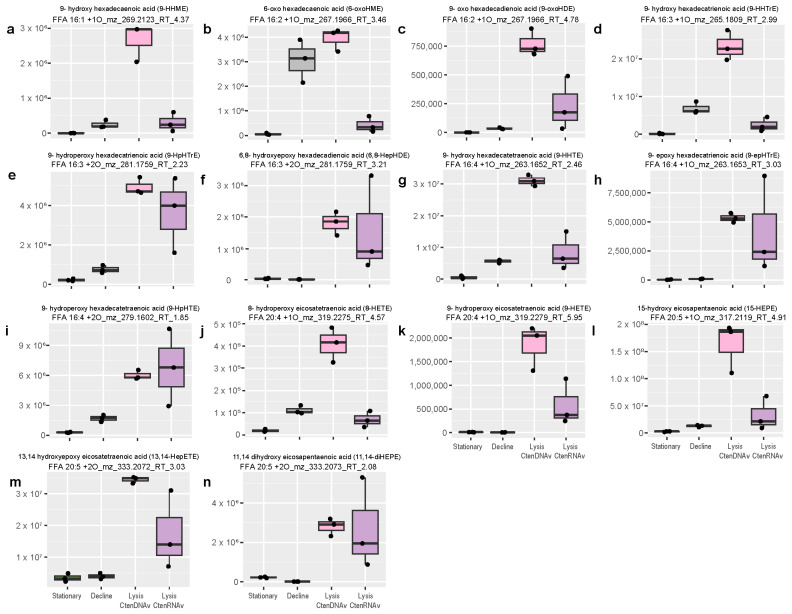
Homologs of allelopathic compounds known to harm both copepods and microzooplankton grazers in field and culture studies were significantly upregulated by viral infection of *Chaetoceros tenuissimus* (ANOVA, Fisher’s LSD post-hoc *p*-value << 0.05 with FDR correction). (**a**–**n**) Box and whisker plots comparing relative abundance of potentially allelopathic compounds that were significantly more abundant during lysis of *C. tenuissimus* by CtenDNAV (light pink) and CtenRNAV (lilac) than in the control treatments during stationary phase (light green) and in decline (grey). Every compound presented was significantly more abundant in the CtenDNAV and CtenRNAV lysis timepoints compared to the control treatments, except for 6-oxoHME (panel (**b**)) and 8-HETE (panel (**j**)). Extracted ion chromatograms for these homologs can be found in Appendix A.

**Figure 5 marinedrugs-22-00228-f005:**
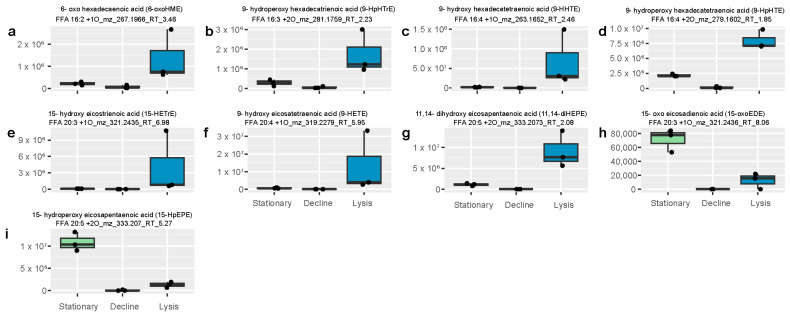
Seven homologs of allelopathic compounds known to harm both copepods and microzooplankton grazers in field and culture studies were significantly upregulated by viral infection of *Chaetoceros socialis* with CsfrRNAV (ANOVA, Tukey post-hoc *p*-value << 0.05 with FDR correction), while two allelopathic compounds were significantly more abundant in the stationary phase lipidome of the host-only control. (**a**–**i**) Box and whisker plots comparing relative abundance of compounds identified as potentially allelopathic oxylipins using ms^2^ fragmentation in the stationary and decline phase controls to lysis treatments. Allelopathic homologs belonging to 6-LOX (**a**), 9-LOX (**b**–**d**,**f**), 11-LOX (**g**), and 15-LOX (**e**,**g**) oxylipin biosynthesis pathways were induced by CsfrRNAV infection. Two 15-LOX C20:5 oxylipins were also associated with stationary growth of the host *C. socialis* (**h**,**i**). Extracted ion chromatograms for these homologs can be found in Appendix A.

**Figure 6 marinedrugs-22-00228-f006:**
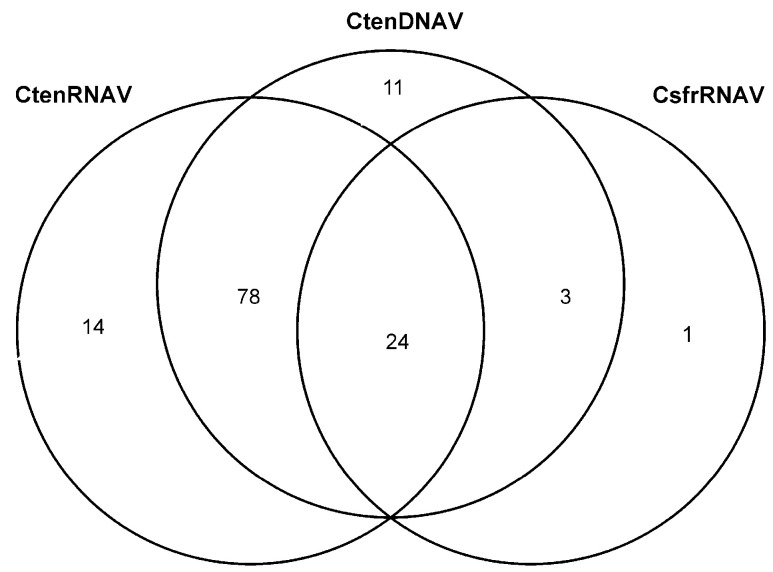
Unique and shared oxylipins associated with viral lysis in three diatom host–virus pairs. A Venn diagram showing the overlap in the significantly upregulated lipidomes (lysis vs. stationary and decline; ANOVA with Fisher’s posthoc *p* < 0.05) of CtenDNAV (*N* = 116), CtenRNAV (*N* = 116), and CsprRNAV (*N* = 28). CtenDNAV infection led to increased production of 11 unique compounds. CtenRNAV infection led to the increased production of 14 unique compounds, and CsfrRNAV led to only one unique compound being overproduced. The most overlap was observed between CtenDNAV and CtenRNAV with 78 shared overproduced compounds. Whereas 24 compounds made up the core oxylipidomic response to viral infection. Only three compounds were shared between CsfrRNAV that were not also overproduced in CtenRNAV.

**Figure 7 marinedrugs-22-00228-f007:**
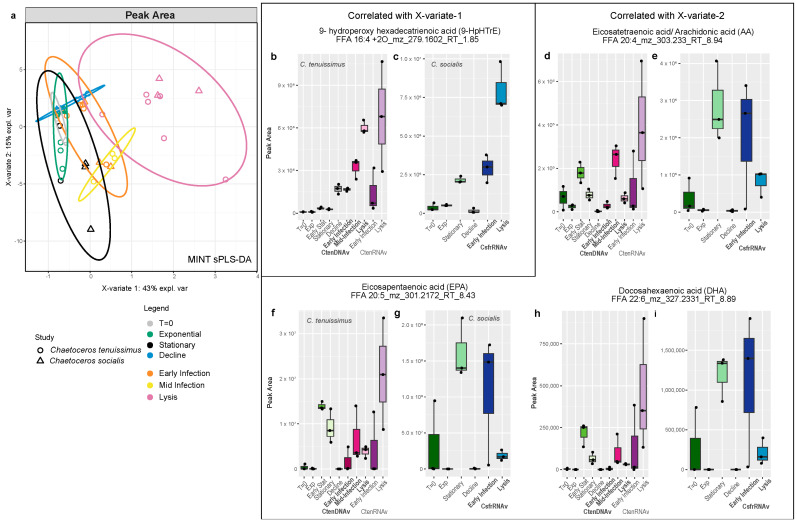
(**a**) Multivariate analysis (sPLSDA) was used to assess the dissolved lipidomes of all of the experiments together. X-variate 1 correlated with lysis and X-variate 2 captures the signal associated with stationary phase. Ellipses show 95% confidence intervals. (**b**,**c**) Box and whisker plot showing the average peak area of 9-HpHTE in the control and virally infected treatments at each timepoint for experiments infecting *Chaetoceros* with CtenDNAV (pink), CtenRNAV (purple), and CsfrRNAV (blue). (**d**–**i**) Box and whisker plot showing the average peak area of (**d**,**e**) arachidonic acid, (**f**,**g**) eicosapentaenoic acid, and (**h**,**i**) docosahexaenoic acid in the control and virally infected treatments at each timepoint for experiments infecting *Chaetoceros* with CtenDNAV (pink), CtenRNAV (purple), and CsfrRNAV (blue).

**Figure 8 marinedrugs-22-00228-f008:**
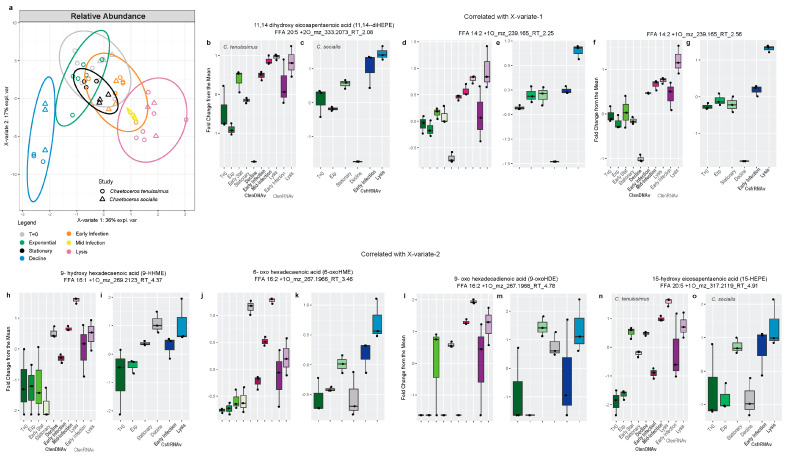
(**a**) Multivariate analysis (sPLS-DA) was used to collectively assess the dissolved lipidomes from all experiments using the relative abundance of each compound to the mean across the lipidome. X-variate 1 correlated with lysis and X-variate 2 captured the signal associated with control decline phase. Ellipses show 95% confidence intervals. (**b**–**g**) Box and whisker plot showing the average peak area of (**b**,**c**) 11,14 dihydroxy eicosapentaenoic acid, (**d**,**e**) FFA 14:2 +1O RT-2.25, and (**f**,**g**) FFA 14:2 +1O RT-2.56 in the control and virally infected treatments at each timepoint for experiments infecting *Chaetoceros* with *CtenDNAV* (pink), *CtenRNAV* (purple), and CsfrRNAV. (**h**–**o**) Box and whisker plot showing the average peak area of known allelopathic oxylipins, (**h**,**i**) 9-HHME, (**j**,**k**) 6-oxoHME, (**l**,**m**) 9-oxoHME, and (**n**,**o**) 15 HEPE in the control and virally infected treatments at each timepoint for experiments infecting *Chaetoceros* with CtenDNAV (pink), CtenRNAV (purple), and CsfrRNAV (blue).

**Figure 9 marinedrugs-22-00228-f009:**
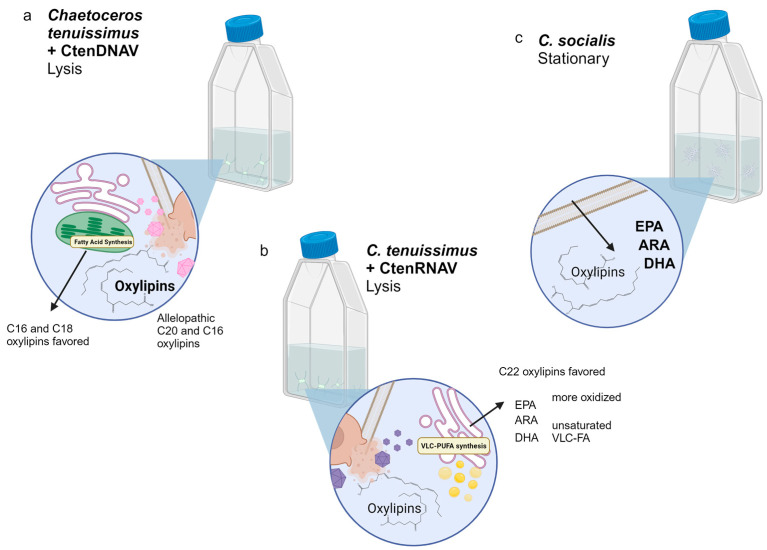
Summary of the unique lipidomic features associated with each diatom host–virus pair during lysis. (**a**) *Chaetoceros tenuissimus* infected with CtenDNAV makes a distinct suite of oxylipins and when compared to CtenRNAV, CtenDNAV makes more allelopathic C20 and C16 compounds, and favors smaller chain length oxylipins with fewer oxygens. These lipids are hypothesized to come from the chloroplast (green plastid) where C16 and C18 fatty acids are synthesized. (**b**) *C. tenuissimus* infected with CtenRNAV produced more C22:n oxylipins with more oxygens than Ct lysed by CtenDNAV, as well as unoxidized very long chain polyunsaturated fatty acids (VLC-PUFAs) like ARA, EPA, and DHA and somewhat rare unsaturated very long chain fatty acids (VLC-FA). Fatty acid elongation and desaturation occurs primarily in the endoplasmic reticulum (pink folded membranes) and VLC-PUFAs are stored in lipid droplets as triacylglycerides (yellow spheres). (**c**) *C. socialis* in stationary phase produced significantly more PUFAs than during CsfrRNAV lysis phase. The size of the text for free fatty acids and oxylipins is representative of their relative abundance.

**Table 1 marinedrugs-22-00228-t001:** Categorization of lipidomic samples by host species, growth phase, and stage of infection.

Sample	Timepoint	Classification	*N*
*C. tenuissimus* 2–10	0	T = 0	3
	1 d	Exponential	3
	3 d	(Early) Stationary	3
	4 d	Stationary	3
	26 d	Decline	3
*C. tenuissimus* + CtenDNAV	1 d	Early Infection	3
	3 d	Mid Infection	3
	4 d	Lysis	3
*C. tenuissimus* + CtenRNAV	1 d	Early Infection	3
	3 d	Lysis	3
*C. socialis* L4	0	T = 0	3
	1 d	Exponential	3
	3 d	Stationary	3
	21 d	Decline	3
*C. socialis* + CsfrRNAV	1 d	Early Infection	3
	3 d	Lysis	3

**Table 2 marinedrugs-22-00228-t002:** Non-volatile oxylipins observed in this study that were annotated as compounds correlated with decreased hatching success of copepods in the Mediterranean [18] at either the structural, functional group, or positional level. The diatom species from which each molecule was originally extracted is also noted: *Skeletonema marinoi* (Sm) [39], *Chaetoceros socialis* (Cs) [39], *Chaetoceros affinis* (Ca) [39], *Thalassiosira rotula* (Tr) [40], *Pseudo-nitzchia delicatissima* (Pd) [41], and *Skeletonema pseudocostatum* (Sp) [37]. Four levels of annotation were given for each molecular species. The LOBSTAHS [42] lipidomic pipeline used this study automatically annotates to the structural level. Many of these compounds were annotated to the functional group level using a retention time model and a smaller subset of compounds were annotated to the positional level using ms^2^ fragmentation.

Diatom Species	Chiral Level	Positional Level	This Study	Functional Group Level	This Study	Structural Level	This Study
Sm, Tr	(7*E*)-9-hydroxy-7-hexaenoic acid	9-HHME	x	HHME	x	FFA C16:1 +1O	x
	(7*E*)-9-oxo-7-hexadecaenoic acid	9-oxoHME		oxoHME	x	FFA C16:2 +1O	x
	(6*Z*, 9*S*, 10*E*, 12*Z*)-9-hydroxy-6,10,12-hexadecatrienoic acid	9-HHTrE	x	HHTrE	x	FFA C16:3 +1O	x
Sm	(6*Z*, 9*S*, 10*E*, 12*Z*)-9-hydroperoxy-6,10,12-hexadecatrienoic acid	9-HpHTrE	x	HpHTrE	x	FFA C16:3 +2O	x
	(6*Z*, 9*S*, 10*E*, 12*Z*, 15*E*)-9-hydroperoxy-6,10,12,15-hexadecatetraenoic acid	9-HpHTE	x	HpHTE	x	FFA C16:4 +2O	x
Sm	(6*Z*, 9*RS*, 10*SR*, 11*SR*, 12*Z*)-11-hydroxy-9,10-epoxyhexadeca-6,12-dienoic acid	11,9-HepHDE		HHDE	x	FFA C16:3 +2O	x
Sm, Pd, Sp	(5*Z*, 8*Z*, 11*Z*, 13*E*, 15S, 17*Z*)-15-hydroxy-5,8,11,13,17-eicosapentaenoic acid	15-HEPE	x	HEPE	x	FFA C20:5 +1O	x
Sm	(5*Z*, 8*Z*, 11*Z*, 13*E*, 15*S*, 17*Z*)-15-hydroperoxy-5,8,11,13,17-eicosapentaenoic acid	15-HpEPE	x	HpEPE	x	FFA C20:5 +2O	x
Sm	(5*R*, 6*E*, 8*Z*, 11*Z*, 14*Z*, 17*Z*)-5-hydroxy-6,8,11,14,17-eicosapentaenoic acid	5-HEPE		HEPE	x	FFA C20:5 +1O	x
Sm	(5*R*, 6*E*, 8*Z*, 11*Z*, 14*Z*, 17*Z*)-5-hydroperoxy-6,8,11,14,17-eicosapentaenoic acid	5-HpEPE		HpEPE	x	FFA C20:5 +2O	x
Sm, Tr, Pd, Sc	(5*Z*, 8*Z*, 11*Z*, 13*RS*, 14*RS*, 15*SR*, 17*Z*)-13-hydroxy-14,15-epoxyeicosa-5,8,11,17-tetraenoic acid	13,14-HepETE	x	HepETE	n.a.	FFA C20:5 +2O	x
Sm	(5*Z*, 7*E*, 9*S*, 11*Z*, 14*Z*, 17*Z*)-9-hydroxy-5,7,11,14,17-eicosapentaenoic acid	9-HEPE		HEPE	x	FFA C20:5 +1O	x
Sm	(5*Z*, 7*E*, 9*S*, 11*Z*, 14*Z*, 17*Z*)-9-hydroperoxy-5,7,11,14,17-eicosapentaenoic acid	9-HpEPE		HpEPE	x	FFA C20:5 +2O	x
Cs	(5*Z*, 7*SR*, 8*RS*, 9*SR*, 11*Z*, 14*Z*, 17*Z*)-7-hydroxy-8,9-epoxy-5,11,14,17-eicosatetraenoic acid	7,8-HepETE		HepETE	n.a.	FFA C20:5 +2O	x
Cs	(5*Z*, 8*Z*, 11*Z*, 15*E*, 17*Z*)-14-hydroxy-(5,8,11,15,17)-eicosapentaenoic acid	14-HEPE		HEPE	x	FFA C20:5 +1O	x
Ca	(5*Z*, 8*Z*, 11*Z*, 15*E*, 17*Z*)-14-hydroperoxy-(5,8,11,15,17)-eicosapentaenoic acid	14-HpEPE		HpEPE	x	FFA C20:5 +2O	x
Ca	(5*Z*, 8*Z*, 11*Z*, 14*SR*, 15*RS*, 16*SR*, 17*Z*)-16-hydroxy-14,15-epoxy-5,8,11,17-eicosatetraenoic acid	16,14-HepETE		HepETE	n.a.	FFA C20:5 +2O	x

**Table 3 marinedrugs-22-00228-t003:** Core oxylipins of viral infection of *Chaetoceros* spp. Twenty-four compounds that were significantly produced during lysis by all three diatom host–virus systems tested. Each compound is represented by their LOBSTAHs annotation, mass-to-charge ratio, and retention time. Seven oxylipins were further annotated to the functional group and positional level using ms^2^ fragmentation. These fully annotated oxylipins are thought to be allelopathic to zooplankton.

LOBSTAHS Annotation	Positional-Level Annotation
FFA_ 14:2 +1O_mz_239.165_RT_1.71	
FFA_ 14:2 +1O_mz_239.165_RT_2.25	
FFA_ 14:2 +1O_mz_239.165_RT_2.56	
FFA_ 16:2 +1O_mz_267.1966_RT_3.46	6-oxoHME
FFA_ 16:3 +2O_mz_281.1759_RT_2.23	9-HpHTrE
FFA_ 16:3 +3O_mz_297.1707_RT_1.52	
FFA_ 16:3 +3O_mz_297.1708_RT_2.13	
FFA_ 16:4 +1O_mz_263.1652_RT_2.46	9-HHTE
FFA_ 16:4 +2O_mz_279.1602_RT_1.85	9-HpHTE
FFA_ 20:1 +1O_mz_325.2749_RT_8.57	
FFA_ 20:2 +1O_mz_323.2593_RT_7.89	
FFA_ 20:3 +1O_mz_321.2435_RT_6.98	15-HETrE
FFA_ 20:3 +3O_mz_353.2335_RT_2.14	
FFA_ 20:4 +1O_mz_319.2279_RT_5.95	9-HETE
FFA_ 20:4 +4O_mz_367.2126_RT_1.58	
FFA_ 20:5 +2O_mz_333.2073_RT_2.08	11,14-diHEPE
FFA_ 22:4 +1O_mz_347.2593_RT_7.77	
FFA_ 22:4 +3O_mz_379.2492_RT_2.47	
FFA_ 22:5 +1O_mz_345.2436_RT_7.12	
FFA_ 22:5 +1O_mz_345.2436_RT_7.67	
FFA_ 22:5 +2O_mz_361.2388_RT_2.91	
FFA_ 22:5 +3O_mz_377.2335_RT_2.21	
FFA_ 22:5 +4O_mz_393.2286_RT_3.65	
FFA_ 22:6 +3O_mz_375.2173_RT_1.72	

## Data Availability

Raw lipidome files were deposited into the MassIVE repository. https://doi.org/10.25345/C5X921W38 (accessed on 1 May 2024).

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
