# Peer review of "Viral Infection Leads to a Unique Suite of Allelopathic Chemical Signals in Three Diatom Host–Virus Pairs"

_marinedrugs, 2024, doi:10.3390/md22050228_

Round 1

Reviewer 1 Report

Comments and Suggestions for Authors

The manuscript presents a comprehensive investigation into the production of oxylipins as chemical defense compounds in diatoms in response to viral infection. The findings demonstrate that viral infection and subsequent lysis of diatoms lead to increased oxylipin production, with the specific compounds produced varying depending on the diatom host and infecting virus. Furthermore, the study highlights that the oxylipidome mediated by viruses differs significantly from oxylipins produced under stress associated with growth phases. The use of high-resolution mass spectrometry to analyze changes in the dissolved lipidome of infected diatom cells adds strength to the research methodology. By investigating three host-virus pairs, the researchers elucidated species-specific differences in oxylipin production, highlighting the importance of considering viral strain specificity. Notably, certain oxylipins significantly overproduced during viral infection are known to impact copepod reproductive success and interfere with microzooplankton grazing, indicating potential ecological implications. Moreover, the identification of specific biomarkers for Chaetoceros diatom infection and differential responses between viral strains provide valuable insights into the complexity of host-virus interactions.

Overall, this research contributes to our understanding of the ecological consequences of viral infection in marine ecosystems and suggests avenues for future investigation into the interplay between viral infection, chemical defense, and ecosystem dynamics. Therefore, we recommend this manuscript for publication in Marine Drugs after minor revision, as it contributes to the field of marine virology and chemical ecology.

 The main points are as follows:

1. Please indicate the corresponding author's name in the article with an asterisk (*) symbol.

 2. The configurations (R, S, Z, and E) of the compounds described in Table 2 should be indicated in italics. The same issue also exists in some parts of the manuscript.

 3. Please modify the format of Table 3 to match the format of Tables 1 and 2.

 4. In the Results section, there are two identical sections labeled 2.2, designated as 2.2. Defining the oxylipidome and 2.2. Growth Phase and Viral Infection Changes the Dissolved Lipidome, respectively. Please revise accordingly."

 5. Figures 7b-h are not very clear. Could you increase the resolution of Figures 7b-7h?

 6. The reference format in the manuscript is incorrect. Please carefully revise it according to the requirements of the Marine Drugs.

 7. "Besides high-resolution accurate mass spectrometry, are there any other techniques available to observe changes in the dissolved lipidome of diatom cells?

 8. Do the oxylipins produced by diatoms inhibit the growth of viruses? What are the biotechnological applications of these oxylipins?

Comments on the Quality of English Language

good

Author Response

Thank you for your thoughtful comments and edits which we have incorporated into the manuscript.

The main points are as follows:

  1. Please indicate the corresponding author's name in the article with an asterisk (*) symbol.

Thank you. This has been corrected.

  1. The configurations (R, S, Z, and E) of the compounds described in Table 2 should be indicated in italics. The same issue also exists in some parts of the manuscript.

Thank you. This has been corrected.

  1. Please modify the format of Table 3 to match the format of Tables 1 and 2.

Thank you. This has been corrected.

  1. In the Results section, there are two identical sections labeled 2.2, designated as 2.2. Defining the oxylipidome and 2.2. Growth Phase and Viral Infection Changes the Dissolved Lipidome, respectively. Please revise accordingly."

 Thank you. This has been corrected.

  1. Figures 7b-h are not very clear. Could you increase the resolution of Figures 7b-7h?

 Yes, I have updated the figure.

  1. The reference format in the manuscript is incorrect. Please carefully revise it according to the requirements of the Marine Drugs.

Thank you. This has been corrected.

  1. "Besides high-resolution accurate mass spectrometry, are there any other techniques available to observe changes in the dissolved lipidome of diatom cells?

 Yes, there are several other methods for measuring oxylipins including different types of mass spectrometry, various derivatization schemes, and alternate annotation pipelines. I’ve cited a number of these methods in the methods section as we did not expound upon our choice of methodology earlier in the text (line 725-728).  

  1. Do the oxylipins produced by diatoms inhibit the growth of viruses? What are the biotechnological applications of these oxylipins?

We did not test whether adding exogenous oxylipins to infected cultures changed the course of viral infection. But this would be an interesting experiment to run. I added a sentence referring to review papers on the commercial value of oxylipins given the antimicrobial, anticancer properties of PUAs (line 540-542).

Reviewer 2 Report

Comments and Suggestions for Authors

The MS” Viral infection leads to a unique suite of allelopathic chemical  signals in three diatom host -virus pairs” that indicates viral infection of diatoms elicits produce diverse oxylipins with downstream trophic and biogeochemical effects

Comments

-Why the author specify the Three host virus pairs :Chaetoceros tenuissimus infected with 19 CtenDNAV; C. tenuissimus infected with CtenRNAV; and Chaetoceros socialis infected with CsfrR- 20 NAV as model systems?

-More details need to be added to the method section

-Viral and diatom propagation conditions

- viral lysis and aging how been performed to induced the release of free fatty acids and oxylipins.

- Some minor points some names need to be italic, abbreviation need to be explained at first appearance

Comments on the Quality of English Language

The MS” Viral infection leads to a unique suite of allelopathic chemical  signals in three diatom host -virus pairs” that indicates viral infection of diatoms elicits produce diverse oxylipins with downstream trophic and biogeochemical effects

Comments

-Why the author specify the Three host virus pairs :Chaetoceros tenuissimus infected with 19 CtenDNAV; C. tenuissimus infected with CtenRNAV; and Chaetoceros socialis infected with CsfrR- 20 NAV as model systems?

-More details need to be added to the method section

-Viral and diatom propagation conditions

- viral lysis and aging how been performed to induced the release of free fatty acids and oxylipins.

- Some minor points some names need to be italic, abbreviation need to be explained at first appearance

Author Response

Thank you for your comments. We have addressed them to the best of our ability. 

Comments

-Why the author specify the Three host virus pairs :Chaetoceros tenuissimus infected with 19 CtenDNAV; C. tenuissimus infected with CtenRNAV; and Chaetoceros socialis infected with CsfrR- 20 NAV as model systems?

These host-virus pairs were chosen because they are one of the very few diatom-virus systems in culture, and Chaetoceros is a dominant genus of diatoms in the ocean. We refer to them as a model systems because we are using them as such, a laboratory model for viral infection of diatoms.

-More details need to be added to the method section

-Viral and diatom propagation conditions

I have added more details about the culturing conditions to the methods section (lines 658-659)

- viral lysis and aging how been performed to induced the release of free fatty acids and oxylipins.

We only filtered out the cell and extracted oxylipins from the filtrate. We did not do anything to further elicit the production of oxylipins, such as sonication.

- Some minor points some names need to be italic, abbreviation need to be explained at first appearance

Thank you. This has been fixed through out.